# Biological Control of Fruit Rot and Anthracnose of Postharvest Mango by Antagonistic Yeasts from Economic Crops Leaves

**DOI:** 10.3390/microorganisms8030317

**Published:** 2020-02-25

**Authors:** Wilasinee Konsue, Tida Dethoup, Savitree Limtong

**Affiliations:** 1Department of Microbiology, Faculty of Science, Kasetsart University, Bangkok 10900, Thailand; wilasinee.kons@ku.th; 2Department of Plant Pathology, Faculty of Agriculture, Kasetsart University, Bangkok 10900, Thailand; agrtdd@ku.ac.th; 3Academy of Science, The Royal Society of Thailand, Bangkok 10300, Thailand

**Keywords:** biological control, *Lasiodiplodia theobromae*, *Colletotrichum gloeosporioides*

## Abstract

To select antagonistic yeasts for the control of fruit rot caused by *Lasiodiplodia*
*theobromae* and anthracnose caused by *Colletotrichum gloeosporioides* in postharvest mango fruit, 307 yeast strains isolated from plant leaves were evaluated for their antagonistic activities against these two fungal pathogens in vitro. *Torulaspora indica* DMKU-RP31, *T. indica* DMKU-RP35 and *Pseudozyma hubeiensis* YE-21 were found to inhibit the growth of *L. theobromae* whereas only *Papiliotrema aspenensis* DMKU-SP67 inhibited the growth of *C. gloeosporioides*. Antagonistic mechanisms of these four antagonistic yeasts in vitro consisted of the production of antifungal volatile organic compounds (VOCs), biofilm formation and siderophore production. *T. indica* DMKU-RP35 was the most effective strain in controlling fruit rot on postharvest mango fruits. Its action was comparable to that of the fungicide, benomyl, reducing the disease severity by 82.4%, whereas benomyl revealed 87.5% reduction. *P. aspenensis* DMKU-SP67 reduced anthracnose severity by 94.1%, which was comparable to that of using benomyl (93.9%). The antifungal VOCs produced by these yeast strains also reduced the severity of these diseases on postharvest mango fruits but at lower rates than using yeast cells. Therefore, these antagonistic yeasts have the potential for use as biological control agents for the control of fruit rot and anthracnose diseases.

## 1. Introduction

Mango (*Mangifera indica* Linn.) is a commercially important fruit in several tropical and subtropical countries. Thailand was the third largest mango producer worldwide in 2018, next to India and China (FAO, 2018). Annually, Thailand produces over four million tons of mango and the export value is approximately 63.9 million USD; therefore, mango is one of the major cash crops of Thailand (OAE, 2018). The most popular mango cultivar is Nam Dok Mai and is consumed domestically as well as exported; this cultivar originated in Thailand and has fruit with a very sweet taste and a good aroma. Mango fruits are commonly highly susceptible to diseases caused by pathogenic microorganisms because they are rich in water and nutrients that provide an ideal substrate for the development of pathogenic microorganisms. This leads to a reduction in the quality of mango fruit, especially during the postharvest period, and causes economic losses [1,2]. The major causes of mango fruit losses are postharvest diseases, including fruit rot (stem-end rot) disease caused by *Lasiodiplodia theobromae* and anthracnose caused by *Colletotrichum gloeosporioides* [3,4]. The symptoms of fruit rot caused by *L. theobromae* appear as soft brown to black lesions at the stem-end of the fruit. The fungus colonizes the surface of the immature fruit as a latent infection and shows no symptoms before harvest but develops after harvest under high relative humidity and high temperature (over 30 °C). The disease symptoms start from the stem end of the fruit and spread quickly through the whole fruit. Moreover, the fungus also infects the fruit through lesions which enlarge and become brown to black [5]. The disease symptoms of anthracnose caused by *C. gloeosporioides* appear as dark brown and black lesions. The fungus infects mango fruit by producing appressoria from germinating spores that penetrate the surface of the fruits. In immature fruit, the fungus usually remains dormant until the fruit begins to ripen. Upon ripening, dark spots develop, enlarging to form lesions which may coalesce to cover almost the entire surface of the fruit [6].

The common strategy for controlling postharvest diseases is using chemical fungicides with high efficiency against fungal pathogens. Such fungicides including prochloraz, carbendazim, benomyl, and thiabendazole are generally used to control fungal pathogens by spraying the postharvest mango fruits before wrapping and/or dipping the fruits in fungicide solution [7]. However, due to growing concerns about the potential risk fungicides pose for human health, environmental contamination, and the development of fungicide resistance by pathogens [8], a control strategy based on antagonistic microorganisms, or biocontrol, has become an attractive alternative approach. Bacteria, yeasts and filamentous fungi are considered to be potential biocontrol agents with broad-spectrum antifungal activities. Among potential antagonistic microorganisms, yeasts deserve particular attention as they are non-pathogenic microorganisms, have a simple nutritional requirement, do not produce allergenic spores, mycotoxins or secondary metabolites as many mycelial fungi do, and can grow rapidly and colonize a broad range of substrates under a wide variety of conditions for long periods of time. Moreover, yeasts are able to adapt to the fruit environment including high sugar concentration and high osmotic pressure [9,10,11]. At present, some yeast species have been reported to control *C. gloeosporioides*, such as *Debaryomyces nepalensis,* which was isolated from soil in China and showed the ability to control the growth of *C. gloeosporioides* [12]. *Pichia membranaefaciens* has the ability to reduce anthracnose incidence in citrus fruit caused by *C. gloeosporioides* [13]. In Mexico, the antagonistic yeasts isolated from the surface of mango fruits, *Meyerozyma caribbica* and *Cryptococcus laurentii*, were evaluated for effectiveness against *C. gloeosporioides* in vitro and showed high potential in reducing the incidence of anthracnose disease caused by *C. gloeosporioides* in mango fruit [14,15]. Fewer studies have dealt with the use of antagonistic yeasts to control *L. theobromae*. *Saccharomyces cerevisiae, Cystobasidium calyptogenae* and *Pichia kudriavzevii* were found to effectively inhibit *L. theobromae* causing fruit rot (stem-end rot) in mango in vitro [16]. *Pichia anomala* was reported to effectively reduce growth of *L. theobromae* in vitro on guava fruit [17].

Antagonistic yeasts display several possible mechanisms against fungal pathogens, for instance, competition for nutrients (e.g., carbohydrate, nitrogen, oxygen) and space through biofilm formation by antagonistic yeast, release of hydrolytic enzymes, parasitism, production of antifungal volatile organic compounds (VOCs), and stimulation of host defense pathways (induction of host resistance) [8,11].

In Thailand, only a few studies on the biological control of postharvest diseases by antagonistic yeasts have been reported. Therefore, the objectives of the present study were to select antagonistic yeasts from yeast strains isolated from plant leaves for controlling of fruit rot and anthracnose diseases of postharvest mango fruits. The selected antagonistic yeast strains were evaluated for their antagonistic mechanism in vitro and efficacy in controlling of fruit rot and anthracnose in postharvest mango fruits.

## 2. Materials and Methods

### 2.1. Microorganisms and Cultivation

The 307 yeast strains used in this studied were isolated from leaves of economic plants (rice, corn and sugarcane) in Thailand [18,19,20] and consisted of 124 strains of 32 ascomycetous species and 178 strains of 15 basidiomycetous species (Appendix A). To obtain active cultures, yeasts were grown on yeast extract-malt extract agar (YM, 3 g/L yeast extract, 3 g/L malt extract, 5 g/L peptone, 10 g/L glucose and 15 g/L agar) at 25 °C for 48 h.

*L. theobromae* and *C. gloeosporioides*, the fungi causing fruit rot and anthracnose disease of postharvest mango fruits, respectively, were obtained from the Department of Plant Pathology, Faculty of Agriculture, Kasetsart University, Bangkok, Thailand. To obtain active cultures, *L. theobromae* and *C. gloeosporioides* were grown for 3 days and 14 days, respectively, on potato dextrose agar (PDA; 4 g/L of potato infusion, 20 g/L glucose and 15 g/L agar) at 25 °C.

### 2.2. Selection of Antagonistic Yeasts Capable of Inhibiting Fungal Pathogens Causing Postharvest Mango Fruit Diseases

The evaluation of antagonistic activities of the 307 yeast strains against the two fungal pathogens causing postharvest diseases of mango fruit was carried out by dual cultivation of a yeast and a fungal pathogen on a PDA plate [21]. The PDA plate was inoculated by linear streaking with a loop of active yeast cells at about 3 cm away from one edge of the plate. After incubation of the plate at 25 °C for 48 h, a 4-mm diameter disk of an actively growing fungal pathogen was inoculated at the opposite edge of the plate at about 3 cm away from the yeast streak. Plates were incubated at 25 °C for 3 days for *L. theobromae* and for 14 days for *C. gloeosporioides*. The control was a plate inoculated with only the fungal pathogen. Three replicates were performed for each treatment. Antagonistic activity was indicated by the presence of an inhibition zone. The percentage of inhibition of mycelium growth was calculated according to the formula {(R_1_ − R_2_)/R_1_} × 100, where R_1_ was the radius of the fungal mycelium growth in the control treatment and R_2_ was the radius of the mycelium growth of the fungus that was exposed to the antagonistic yeast.

### 2.3. Determination of Antagonistic Mechanisms of Antagonistic Yeasts in Vitro

#### 2.3.1. Production of Antifungal Volatile Organic Compounds

To evaluate the production of antifungal VOCs by the antagonistic yeasts, the double plate assay method of Francesco et al. [22] was used with slight modifications. A cell suspension of an antagonistic yeast was prepared by suspending a 2-day-old culture in sterile 0.85% NaCl at a concentration of 10^8^ cells/mL. An aliquot (100 µL) of the yeast cell suspension was spread on a YM agar plate and incubated at 25 °C for 2 days. After incubation, the cover of the plate was removed and the plate was inverted and placed upside down on the bottom of another PDA plate that was inoculated with a 4-mm diameter disk of an actively growing fungal pathogen placed in the middle of the plate. The two bottom plates were sealed with a double layer of parafilm and incubated at 25 °C for 3 days for *L. theobromae* and 14 days for *C. gloeosporioides*. The control treatment was a PDA plate inoculated with only a disk of the actively growing fungal pathogen. Each treatment was carried out with three replicates. The percentage of inhibition of fungal growth by VOCs produced by the antagonistic yeast was calculated from the same formula, {(R_1_ − R_2_)/R_1_} × 100.

#### 2.3.2. Competition for Nutrients

Competition for nutrients was estimated from the inhibition of fungal pathogen growth by a yeast on media containing different of nutrient concentrations using the dual cultivation method. PDA plates were prepared following Zhang et al. [23] with four different nutrient concentrations: normal nutrient concentration (39 g/L PDA powder), half of the normal nutrient concentration (19.5 g/L PDA powder), one-fourth of the normal of nutrient concentration (9.7 g/L PDA powder) and one-tenth of the normal of nutrient concentration (3.9 g/L PDA powder).

#### 2.3.3. Siderophore Production

Evaluation of siderophore production by the antagonistic yeasts was performed by the method of Louden et al. [24] using chrome azurol S (CAS)-blue agar with slight modifications. A cell suspension of an antagonistic yeast (10 µL) was prepared as previously described, and the cell suspension was dropped on the CAS-blue agar plate and incubated at 25 °C in the dark for 10 days. The negative control was a CAS-blue agar plate on which only 0.85% NaCl was dropped. Each treatment was performed with three replications. Siderophore production was indicated by a change of the color of the CAS-blue agar from blue to yellow or orange.

#### 2.3.4. Biofilm Formation

Evaluation of biofilm formation by the antagonistic yeasts was performed by the method of Ruzicka et al. [25] with slight modification. An aliquot (20 µL) of a yeast cell suspension prepared as previously described was added to each well of a microtiter plate containing 180 µL of potato dextrose broth (PDB) and incubated at 25 °C for 48 h. After incubation, each well was washed thrice with 200 µL of sterile phosphate-buffered saline (PBS; 8 g/L NaCl, 2.9 g/L Na_2_HPO_4_.12H_2_O, 0.2 g/L KH_2_PO_4_ and 0.2 g/L KCl, pH 7.2) to remove all non-adherent cells from the well. After every washing, the well was emptied by flicking the microtiter plate. After the three washings, the adherent cells (biofilm layer) were fixed by air drying. The adherent cells in wells were stained with 200 µL of 2% (w/v) aqueous crystal violet solution for 15 min at room temperature (30 ± 2 °C). The wells were rinsed with distilled water until the washing was free of the stain, and then the stained cells were suspended with 200 µL of 95% ethanol and left at room temperature for 45 min. The optical density (OD) was measured at 600 nm using a microtiter plate reader (Thermo Scientific^TM^, Finland). The PDB without yeast cells was used as the negative control. Each treatment repeated three times. Biofilm formation was considered positive in the wells where the OD value was higher than the cut-off OD value or the OD of the negative control (ODc). The following classification was applied for the determination of biofilm formation: weak biofilm producer (ODc < OD ≤ 2ODc), moderate biofilm producer (2ODc < OD ≤ 4ODc) and strong biofilm producer (4ODc < OD) [26].

### 2.4. Evaluation of the Efficacy of Antagonistic Yeasts in Controlling of Postharvest Diseases of Mango on Mango Fruit

Control of fruit rot (caused by *L. theobromae*) and anthracnose (caused by *C. gloeosporioides*) on postharvest mango fruits by the antagonistic yeasts was evaluated by the method of Bautista-Rosales et al. [14]. The chemical fungicide, 0.1 mg/mL benomyl, was used for comparison. Fresh, ripe mango fruits with no sign of lesions were acquired from a farm in Nonthaburi province. The mango fruits were surface disinfected by dipping into 70% ethanol for 1 min and into 0.5% sodium hypochlorite (NaOCl) for 1 min and then rinsing in sterile water three times. After air drying at room temperature, two 2-mm deep wounds were made in each fruit with a sterile needle and 20 microliters of 2-day-old yeast cell suspension (10^8^ cell/mL), which had been prepared as described previously, was inoculated into each wound. After 1 h, 20 µL of a conidial suspension (10^5^ spores/mL) of fungal pathogen, which had been prepared by mixing conidia from a 3-day-old *L. theobromae* culture or a 14-day-old *C. gloeosporioides* culture grown on PDA at 25 °C with sterile 0.01% tween 80, was applied to each wound. In the positive control treatment, sterile 0.85% NaCl was substituted for the yeast cell suspension before application of the spore suspension of fungal pathogen. Whereas in the negative control treatment, only sterile 0.85% NaCl (40 µL) was applied to the wound. Each treatment consisted of a group of five fruits. All fruits were transferred to a sterile plastic chamber and incubated at 25 °C and high relative humidity for 7 days. The inoculated mango fruits were evaluated by measuring the diameter of the lesions. The disease severity was calculated as described by Nadai et al. [27] with the formula {(D_1_ − D_2_)/D_1_} ×100, where D_1_ was the disease lesion diameter on a positive control mango fruit and D_2_ was the lesion diameter of mango fruit of the treatment (fungal pathogen and antagonistic yeast). The experiment was performed twice.

### 2.5. Evaluation of the Efficacy of Antifungal Volatile Organic Compounds Produced by Antagonistic Yeasts in the Control of Postharvest Diseases of Mango on Mango Fruit

To evaluate the efficacy of the VOCs produced by the antagonistic yeasts, the biofumigation method of Francesco et al. [22] was used. Mango fruits were surface disinfected and air dried. Then, two 2-mm deep wounds were made in each mango with a sterile needle and 20 µL of a conidial suspension (10^5^ spores/mL) of fungal pathogen was applied to each wound. The inoculated mango fruits were transferred to a sterile desiccator in the bottom of which a YM agar plate inoculated with 2-day-old yeast cell suspension (10^8^ cell/mL) had been placed. The inoculated fruits were placed on a perforated ceramic plate to prevent direct contact between the fruits and the yeast culture. The positive control treatment was incubating fruits inoculated with a fungal pathogen in a sterile desiccator alone. The desiccators were closed and sealed, and the fruits were incubated at 25 °C for 7 days. Each treatment consisted of a group of three fruits. The percentage of the disease severity reduction by VOCs was calculated with the formula described above. The experiment was performed twice.

### 2.6. Statistical Analysis

These data were evaluated by analyses of variance (ANOVA) using the SPSS statistics software version 22 for Windows (IBM, New York, USA). Statistical significance was evaluated using Duncan’s multiple range test (DMRT) and a significance level of *p* ≤ 0.05 was considered as being significantly different.

## 3. Results

### 3.1. Selection of Antagonistic Yeasts Capable of Inhibiting Fungal Pathogens Causing Postharvest Mango Fruit Diseases

From among 307 yeast strains evaluated for their antagonistic activity against *L. theobromae* and *C. gloeosporiodes* by dual cultivation, only four strains were able to inhibit the two fungal pathogens. Three yeast strains (*Torulaspora indica* DMKU-RP31, *T. indica* DMKU-RP35 and *Pseudozyma hubeiensis* YE-21) inhibited the growth of *L. theobromae*, which causes fruit rot disease in mango, by 67.9%, 67.7% and 56.7%, respectively (Figure 1, Appendix A), while *Papiliotrema aspenensis* DMKU-SP67 inhibited the growth of *C. gloeosporiodes*, which causes anthracnose in mango, by 66.3% (Figure 1, Appendix A).

### 3.2. Determination of Antagonistic Mechanisms of Antagonistic Yeasts in Vitro

#### 3.2.1. Production of Antifungal Volatile Organic Compounds

The double plate assay was used, in which diffusible compounds from yeast culture did not contact fungal pathogens and inhibition came from the VOCs produced by the antagonistic yeasts. Three antagonistic yeasts (*T. indica* DMKU-RP31, *T. indica* DMKU-RP35 and *Ps. hubeiensis* YE-21) that antagonized *L. theobromae* produced VOCs which inhibited this fungal pathogen by 59.8%, 51.3% and 34.4% respectively (Figure 2; Figure 3, Appendix A). *P. aspenensis* DMKU-SP67, which inhibited *C. gloeosporiodes* growth in dual cultivation, showed the ability to inhibit this fungal pathogen by its VOCs (50.3%) (Figure 2 and Figure 3, Appendix A).

#### 3.2.2. Competition for Nutrients

Dual cultivation of an antagonistic yeast and a fungal pathogen on PDA containing four different nutrient concentrations viz. normal, half of normal, one-fourth of normal and one-tenth of normal PDA concentration was used to determine competition for nutrients. The result revealed that with decreased concentrations of nutrients in PDA medium, all of the antagonistic yeast strains showed higher growth inhibition of the fungal pathogens (Table 1). The inhibition was the highest when the yeast and the fungal pathogen were dual cultured on PDA plate with one-tenth of the normal PDA concentration. The results indicated that there was competition for nutrients.

#### 3.2.3. Siderophore Production

All four antagonistic yeast strains grew well and changed the color of CAS–blue agar from blue to yellow, indicating that they produced siderophores (Figure 4, Table 2). Among these strains, *Ps. hubeiensis* YE-21 showed the largest yellow zone, indicating that it produced the highest amount of siderophores.

#### 3.2.4. Biofilm Formation

Evaluation of biofilm formation by the four antagonistic yeast strains revealed that after 48 h incubation all yeast strains were able to adhere to the well of microtiter plate following three washes. The optical density values at 600 nm (OD_600_) of all strains were found to be higher than those of the negative controls, when the optical density cut-off value was set at OD_600_ = 0.079 (Table 2), indicating that the yeast strains had ability to form biofilms. T. indica DMKU-RP35 was the strongest biofilm producer.

### 3.3. Evaluation the Efficacy of Antagonistic Yeasts in Controlling of Postharvest Diseases of Mango on Mango Fruit

Three antagonistic yeast strains (*T. indica* DMKU-RP31, *T. indica* DMKU-RP35 and *Ps. hubeiensis* YE-21), which inhibited the growth of *L. theobromae*, the cause of fruit rot disease of mango *in vitro*, were evaluated for controlling fruit rot disease on mango fruits. Fresh, ripe mango fruits were inoculated with a cell suspension of each antagonistic yeast strain or the commercial chemical fungicide, benomyl and a conidial suspension of *L. theobromae*. *T. indica* DMKU-RP35 revealed the highest reduction of the disease severity, 82.4%, whereas *T. indica* DMKU-RP31 and *Ps. hubeiensis* YE-21 reduced severity by 49.8% and 42.5%, respectively (Figure 5, Table 3). On the other hand, benomyl reduced disease severity by 87.5%, which was not significantly different when compared with *T. indica* DMKU-RP35.

*P. aspenensis* DMKU-SP67, which inhibited growth of *C. gloeosporiodes* in vitro, was evaluated for control of anthracnose on mango fruits compared with benomyl. The result revealed that *P. aspenensis* DMKU-SP67 reduced disease severity by 94.1% (Figure 5, Table 3), which was not significantly different from benomyl which reduced disease severity by 93.9%.

### 3.4. Evaluation the Efficacy of Antifungal Volatile Organic Compounds Produced by Antagonistic Yeasts in Controlling of Postharvest Diseases of Mango on Mango Fruit

Evaluation of the efficacy of the antifungal VOCs produced by the three antagonistic yeasts (*T. indica* DMKU-RP31, *T. indica* DMKU-RP35 and *Ps. hubeiensis* YE-21) for the control of fruit rot on mango fruits was performed by biofumigation. The result revealed the same trend as in control by using yeast cell suspension. *T. indica* DMKU-RP35 produced the highest reduction in fruit rot severity: 53.1%, while *T. indica* DMKU-RP31 and *Ps. hubeiensis* YE-21 showed relatively low reduction of disease severity: 31.5% and 19.2%, respectively (Table 3). Antifungal VOCs produced by *P. aspenensis* DMKU-SP67 reduced anthracnose severity on mango fruits by 48.5%.

## 4. Discussion

There have been several studies on using antagonistic yeasts for controlling plant diseases or postharvest diseases caused by pathogenic microorganisms, especially fungi [23,28,29]. In this study we focused on using antagonistic yeast strains for controlling postharvest diseases of mango fruit, namely fruit rot caused by *L. theobromae* and anthracnose caused by *C. gloeosporioides.* Effective antagonistic yeast strains were obtained by screening yeast strains isolated from the surface and tissue of leaves of economic crops in Thailand. These yeast strains used in the present study were obtained from our previous studies on the diversity of yeasts in the phylloplane and tissue of economic crops (rice, corn and sugarcane) leaves in Thailand by culture-depend method and some of them revealed antagonistic activities against plant pathogens [18,19,20]. In most cases, yeasts showing potential for controlling of fungal pathogens of one kind of postharvest fruit were obtained from the other fruits or the other sources. For example, *Pichia guilliermondii* isolated from the rhizosphere of corn showed efficacy against *Botrytis cinerea* on apple fruits [30]. *Leucosporidium scottii* obtained from soil was found to be a good biocontrol agent of blue mold caused by *Penicillium expansum* and gray mold caused by *B. cinerea* of two apple cultivars [31]. Two epiphytic yeast strains isolated from the surface of a mango and an orange showed potential to reduce postharvest *Penicillium digitatum* decay on kinnow fruit [32]. The results revealed that only four yeast strains out of 307 strains (1.3%) isolated from plant leaves were capable of inhibiting these two fungal pathogens causing mango fruit disease. Of the strains *T. indica* DMKU-RP31, *T. indica* DMKU-RP35 and *Ps. hubeiensis* YE-21, which effectively inhibited the growth of *L. theobromae*, the first two were isolated from the phylloplane of different rice leaf samples while the third was isolated from the phylloplane of a corn leaves sample. Whereas *P. aspenensis* DMKU-SP67, which inhibited *C. gloeosporioides* growth, was obtained from the phylloplane of a sugarcane leaf sample. The results of previous investigations revealed that some yeast species have been reported to inhibit growth of these two fungal pathogens. For example, *Candida membranifaciens* was able to inhibit spore germination and hyphal growth of *C. gloeosporioides,* the cause of anthracnose on mango fruit [33]. *Meyerozyma guilliermondii* reduced the fungal hyphal growth of *C. gloeosporioides,* the cause of anthracnose on papaya [34]. *Wickerhamomyces anomalus* isolated from avocado leaves showed in vitro antagonistic activity against *C. gloeosporioides*, the cause of anthracnose on avocado [35]. Among four antagonistic yeast strains, *T. indica* DMKU-RP31 and *T. indica* DMKU-RP35 revealed the ability to inhibit plant pathogens’ cause of rice diseases, namely *Curvularia lunata* (cause of dirty panicle disease)*, Fusarium moniliforme* (cause of bakanae disease), *Helminthosporium oryzae* (cause of brown spot disease), *Rhizoctonia solani* (cause of sheath blight disease) and *Pyricularia oryzae* (cause of blast disease) (data not showed). Our results indicated that the antagonistic mechanisms in vitro of these four antagonistic yeast strains against the two postharvest fungal pathogens were related to antifungal VOC production, competition for nutrients, biofilm formation and siderophore production, however to different degrees among yeast strains. Antifungal VOCs are low molecular weight compounds (< 300 Da) with low polarity and high vapor pressure [36]. VOCs are considered as ideal antimicrobials and biofumigants because they do not require physical contact between antagonistic microorganisms and pathogens or between antagonistic microorganisms and the product or food commodity [10,37]. Many yeasts and yeast-like fungal species have been reported to produce VOCs that are potential agents for controlling postharvest fungal pathogens. For example, *Aureobasidium pullulans* inhibited the growth of fungal pathogens causing postharvest diseases in apples [22]. *D. nepalensis* produced VOCs inhibiting *C. gloeosporioides* [12]. VOCs produced by *Candida intermedia* and *Sporidiobolus pararoseus* were reported to inhibit the growth of *B. cinereal* [38,39]. The result of our study showed that all antagonistic yeast strains produced VOCs, which was one of the antagonistic mechanisms against the two fungal pathogens. The most effective VOC producer in this study was *T. indica* DMKU-RP35. However, the inhibition of fungal growth by VOCs produced by antagonistic yeasts was lower than that obtained by using yeast cells. This result indicated that other mechanisms were involved in the antagonistic activity of these yeast strains against the two fungal pathogens. Competition for nutrients has been considered to be one of the primary antagonistic mechanisms of yeasts against fungal pathogens [11]. Antagonistic yeasts can grow where nutrients or resources are limited and can use a broader range of substrates better than pathogens [40,41]. In this study, all yeast strains showed the highest inhibition of pathogenic fungal growth at the lowest nutrient concentration, indicating that competition for nutrients is one of the antagonistic mechanisms of these antagonistic yeasts against the postharvest fungal pathogens. Siderophore production, which is a mechanism involving competition for iron, could have a significant role in the biocontrol of postharvest fungal pathogens [11,42]. Siderophores are low molecular weight compounds that are capable of forming tight and stable complexes with ferric iron (Fe^3+^) which serve in transporting iron across cell membranes. Siderophores can be produced by some yeast species. For example, *Rhodotorula glutinis* produced siderophores that improved the biological control of *P. expansum*, the cause of blue rot in apple [43], and *W. anomalus* produces siderophores which are capable of antagonizing *Curvalaria lunata*, a cause of dirty panicle disease of rice [44]. In the present study, all antagonistic yeast strains produced siderophores. Therefore, siderophore production could be one of the antagonistic mechanisms of these yeasts against these two fungal pathogens. Biofilm formation is one of the main features of antagonistic yeasts and is involved in competition for space [45]. Some yeast strains have the ability to produce extracellular matrices that consist of polysaccharides, proteins and nucleic acids. Biofilms improve the ability of yeast strains to adhere to and colonize fruit surfaces [11,46]. Ianiri et al. [47] reported that *S. cerevisiae* had the ability to form biofilm which limited the growth of fungal pathogens. *Pichia fermentans* was found to be a potential biological control agent to control *Monilinia fructicila* by forming biofilm on peach fruit [48]. The result of our study revealed that all antagonistic yeast strains were capable of producing biofilms but to different degrees. Therefore, we assumed that biofilm formation could be one of the antagonistic mechanisms to antagonize the two postharvest fungal pathogens.

The efficacy of *T. indica* DMKU-RP31, *T. indica* DMKU-RP35 and *Ps. hubeiensis* YE-21 in controlling fruit rot of postharvest mango fruit caused by *L. theobromae* on mango fruits was evaluated. *T. indica* DMKU-RP35 was found to be the most effective antagonistic yeast strain against this disease. The reduction in the severity of fruit rot disease by this yeast strain was comparable to that of the chemical fungicide, benomyl, which is usually used to control this postharvest disease. In evaluation of the efficacy of VOCs produced by these yeast strains, *T. indica* DMKU-RP35 showed the highest fruit rot disease reduction although the disease reduction was lower than that resulting from using yeast cells. This revealed that VOCs production was one of the antagonistic mechanisms in vivo contributing to the antagonistic activity of these antagonistic yeasts against *L. theobromae*.

In control of anthracnose on mango fruit caused by *C. gloeosporioides*, the reduction of the disease severity on the fruits achieved by the antagonistic yeast *P. aspenensis* DMKU-SP67 was as high as that of benomyl. The VOCs produced by this yeast strain also reduced the disease severity even though the reduction was lower than that attained by using yeast cells; the same as was found in the evaluation of the three antagonistic yeast strains against *L. theobromae.* Some yeast species such as *S. cerevisiae, Cy. calyptogenae* and *P. kudriavzevii* were previously reported to reduce fruit rot severity on mango fruit [16], whereas *M. caribbica* and *Cr. laurentii* were found to reduce severity of anthracnose on mango fruit [14,15]. However, the efficacy of *T. indica* and *Ps. hubeiensis* to control fruit rot and that of *P. aspenensis* to control anthracnose have never been reported.

In general, during the storage period of postharvest fruits, changing of fruit quality occurred. This included losing weight and firmness, changing texture and composition (total soluble solids that consist of sucrose, glucose, fructose and some acids), and changing color and flavor (contributed by phenolic compounds in fruits) [30,49]. Some researchers reported that the application of biocontrol yeasts revealed the reduction of fruit quality changes. Wei et al. [50] reported that antagonistic yeast *Cr. laurentii* could reduce the decrease of firmness and ascorbic acid on mango fruits. Luo et al. [51] demonstrated that *D. nepalensis* effectively inhibited anthracnose incidence on mango fruit and delayed the decrease of firmness, and total soluble solids, total acid, and ascorbic acid values. Tian et al. [52] determined the effect of the antagonistic yeast *Metschnikowia pulcherrima*, which inhibited the growth of *C. gloeosporioides*, on storage quality of mango and reported that this yeast could inhibit the changes of quality parameters including peel color, firmness, total soluble solids, total acid, and ascorbic acid, and maintain the storage quality of mango fruits. Habiba et al. [32] reported that weight loss was lesser in yeast-treated kinnow fruit than untreated fruit, whereas total soluble solids were increasing when prolonging storage period both in yeast-treated and untreated fruit. In addition, they found increasing amounts of phenolic compounds for up to ten days storage, which then decreased. In the present study, the effect of the biocontrol yeasts on the storage quality of mango fruits were not evaluated. Hence, this aspect needs to be further studied.

Improving the efficacy of antagonistic yeast strains in the controlling of postharvest diseases could be achieved by application of the appropriate mixture of the antagonistic yeast strains, instead of using an individual strain [53]. It was reported that the combination of *P. guilliermondii* and other five yeast strains was more effective in controlling black rot disease of pineapple caused by *Ceratomyces paradoxa* [54]. In addition, integrated management approaches, such as combination of biocontrol yeasts together with physical treatments such as hot air treatment, hot water treatments, and low dose of ultraviolet-C treatment, as well as combination of biocontrol yeasts along with low-dose fungicides and some compounds such as salicylic acid and indole-3-acetic acid, could improve the efficacy in controlling of postharvest disease. The efficacy of antagonistic yeast *Metschnikowia fructicola* in controlling *P. expansum* in apple fruits was increased when used in combination with hot air treatment (40 °C) [55]. Application of *Candida guilliermondii* or *P. membranaefaciens* with hot water treatment (38 °C) showed significantly controlled loquat fruit decay caused by *B. cinerea* [56]. Using *Cr. laurentii* with ultraviolet-C treatment could control tomato decay caused by *B. cinerea* or *Alternaria alternata* [57]. *Leu. scottii* was reported to be a good biocontrol agent for blue (*P. expansum*) and gray (*B. cinerea*) mold of two apples cultivars, and resistant to commonly used fungicides, i.e., iprodione, thiabendazole and imazalil, therefore, application of this biocontrol yeast along with a low-dose of these fungicides was recommended [31]. Biocontrol efficacy of the antagonistic yeast *Cr. laurentii* against blue mold rot caused by *P. expansum* in apple fruit could be enhanced by the addition of indole-3-acetic acid, a plant growth promotor [58]. Addition of salicylic acid could increase the efficacy of antagonistic yeasts *R. glutinis* on controlling postharvest disease of peach caused by gray mold [59]. Therefore, to improve the efficacy of our four antagonistic yeast strains by controlling fruit rot and anthracnose of postharvest mango fruits, these management techniques should be further studied.

To commercialize these biocontrol yeasts, large amounts of cell mass must be produced and these yeast cells have to be formulated and stabilized to endow the products with long shelf lives and low production costs. Melin et al. [60] developed wet (liquid) and dry formulation (fluidized bed drying, lyophilization and vacuum drying) of the biocontrol yeasts *P. anomala* strain J121. They reported that with all formulations developed, yeast cells with shelf lives of at least a few months were obtained and in all formulations the biocontrol activity was retained. Thus, the research to develop formulation of our biocontrol yeast has to be carried out.

In summary, further research efforts that should be carried out will include determination of the effect on storage quality and improvement of the biocontrol efficacy, as well as a study on how to commercialize these biocontrol yeasts.

## 5. Conclusions

To our knowledge, this is the first report on the effective biocontrol activities of *T. indica* and *Ps. hubeiensis* against *L. theobromae*, the cause of fruit rot disease, and of *P. aspenensis* against *C. gloeosporioides*, the cause of anthracnose disease of postharvest mango fruit. These antagonistic yeasts showed the inhibition of fungal pathogen growth in vitro and in vivo. The results presented in this study demonstrate that these antagonistic yeasts had several antagonistic mechanisms including the production of VOCs and the competition for nutrients and space. The results showed that these antagonistic yeasts have a high potential for use as biological agents for the control of postharvest mango diseases caused by major fungal pathogens.

## Figures and Tables

**Figure 1 microorganisms-08-00317-f001:**
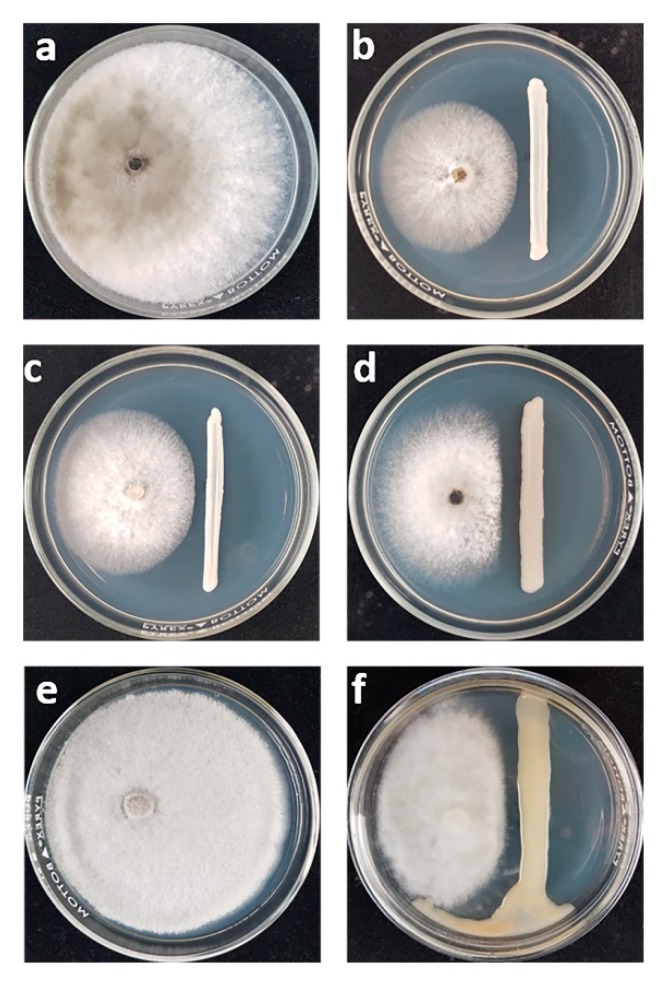
(**a**) Growth on potato dextrose agar (PDA) at 25 °C of *L. theobromae* alone and (**b**) dual cultivation with *T. indica* DMKU-RP31; (**c**) *T. indica* DMKU-RP35; (**d**) *Ps. hubeiensis* YE-21 for three days; (**e**) growth of *C. gloeosporiodes* alone and (**f**) dual cultivation with *P. aspenensis* DMKU-SP67 for 14 days.

**Figure 2 microorganisms-08-00317-f002:**
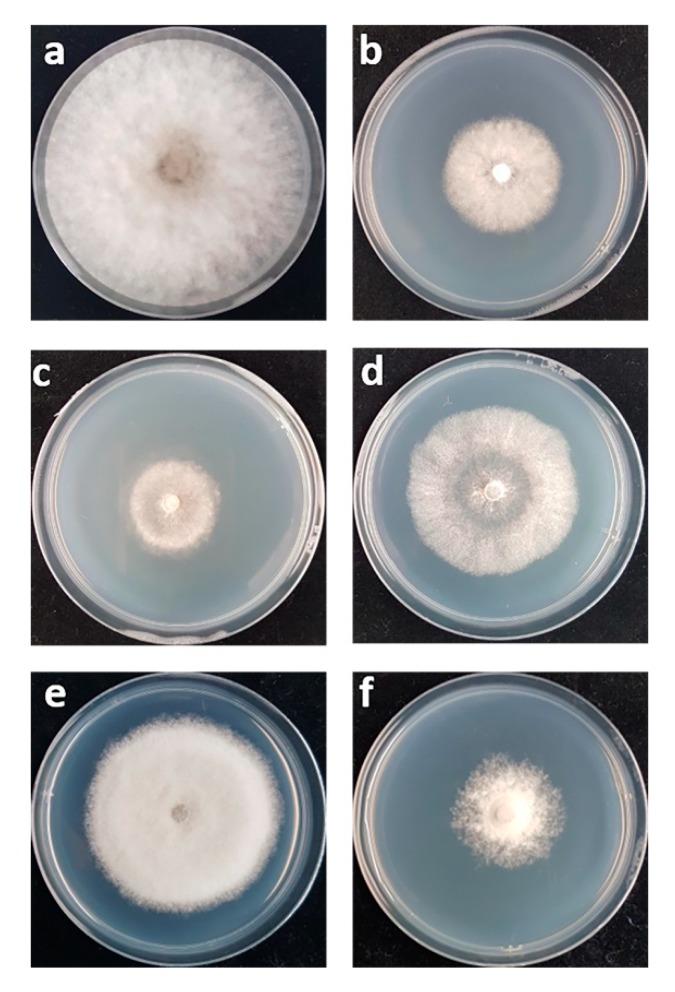
(**a**) Growth on PDA at 25 °C of *L. theobromae* alone and (**b**) with *T. indica* DMKU-RP31; (**c**) *T. indica* DMKU-RP35; (**d**) *Ps. hubeiensis* YE-21 for three days; (**e**) growth of *C. gloeosporiodes* alone and (**f**) dual cultivation with *P. aspenensis* DMKU-SP67 for 14 days by double plates cultivation to determine antifungal volatile organic compounds (VOCs) production.

**Figure 3 microorganisms-08-00317-f003:**
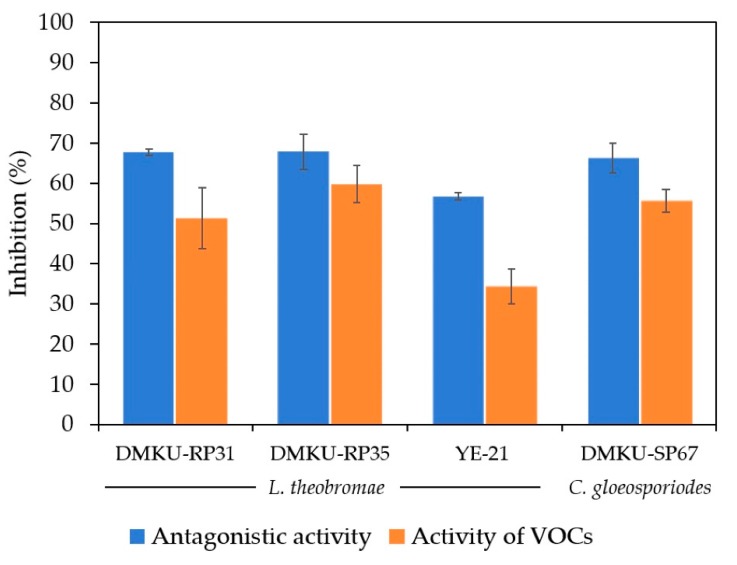
Inhibition of *L. theobromae* and *C. gloeosporiodes* growth on PDA plates at 25 °C for 3 and 14 days, respectively, by antagonistic yeast strains using dual cultivation and VOCs produced by antagonistic yeast strains using double plates.

**Figure 4 microorganisms-08-00317-f004:**
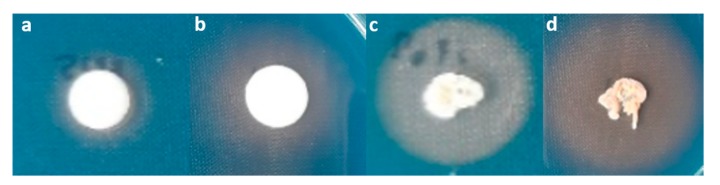
Siderophore production by (**a**) *T. indica* DMKU-RP31; (**b**) *T. indica* DMKU-RP35; (**c**) *Ps. hubeiensis* YE-21; and (**d**) *P. aspenensis* DMKU-SP67 after 10 days of incubation on CAS-blue agars.

**Figure 5 microorganisms-08-00317-f005:**
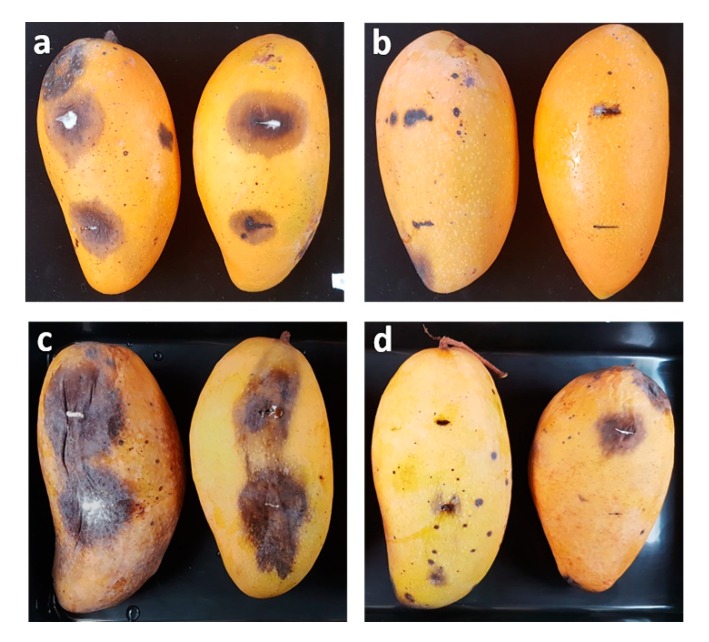
(**a**) Mango fruits inoculated with *C. gloeosporiodes* alone; (**b**) *C. gloeosporiodes* and the antagonistic yeast *P. aspenensis* DMKU-SP67; (**c**) *L. theobromae* alone and (**d**) *L. theobromae* and the antagonistic yeast *T. indica* DMKU-RP35 incubated at 25 °C for 7 days.

**Table 1 microorganisms-08-00317-t001:** Inhibition of *L. theobromae* and *C. gloeosporiodes* by the antagonistic yeast strains on PDA with different nutrient concentrations at 25 °C for 3 and 14 days, respectively.

Treatment	Fungal Pathogens Growth Inhibition (%) ^a^			
	39.0 g/L PDA Powder	19.5 g/L PDA Powder	9.7 g/L PDA Powder	3.9 g/L PDA Powder
*T. indica* DMKU-RP31 + *L. theobromae*	65.6 ± 1.87bc	67.3 ± 0.99b	70.8 ± 1.69ab	75.6 ± 3.24a
*T. indica* DMKU-RP35 + *L. theobromae*	67.7 ± 0.90b	68.6 ± 0.57b	70.8 ± 3.56ab	76.0 ± 4.68a
*Ps. hubeiensis* YE-21 + *L. theobromae*	58.4 ± 1.87d	61.7 ± 1.14cd	62.0 ± 3.56cd	69.7 ± 5.15b
*P. aspenensis* DMKU-SP67 + *C. gloeosporiodes*	64.6 ± 3.72c	68.1 ± 0.57bc	71.3 ± 1.71b	75.7 ± 1.36a

In the same row data followed by the different, same, and overlapping lower-case letters means significantly different, and no significantly different of their overlapping to Duncan’s multiple range test at *p* ≤ 0.05. Each result presents the mean ± standard derivation from three replicates. ^a^ Inhibition (%) = (radius of control fungal colony − radius of fungal colony grows with yeast) × 100/radius of control fungal colony.

**Table 2 microorganisms-08-00317-t002:** Siderophore production and biofilm formation by the antagonistic yeast strains.

Treatment	Siderophore (Yellow Zone Diameter, mm) ^a^	Biofilm Formation		
		OD_600_ (Average ± SD)	OD Value ^b^	Level ^c^
Control ^c^	nd	0.079 ± 0.00	-	-
*T. indica* DMKU-RP31	7.83 ± 0.76	0.192 ± 0.05	2.4303	Moderate
*T. indica* DMKU-RP35	6.33 ± 0.58	0.552 ± 0.03	6.9873	Strong
*Ps. hubeiensis* YE-21	12.67 ± 1.15	0.092 ± 0.01	1.1645	Weak
*P. aspenensis* DMKU-SP67	9.67 ± 0.58	0.103 ± 0.01	1.3037	Weak

nd: not determined. ^a^ Yellow zone on CAS-blue agar. ^b^ Average optical density of samples as a portion of ODc (control). ^c^ The optical density cut-off value (ODc = 0.079) was average OD600 of negative control. Interpretation of biofilm formation: weak biofilm formation (ODc < OD ≤ 2ODc), moderate biofilm formation (2ODc < OD ≤ 4ODc) and strong biofilm formation (4ODc < OD).

**Table 3 microorganisms-08-00317-t003:** Controlling of fruit rot (caused by *L. theobromae*) and the anthracnose (caused by *C. gloeosporiodes)* by cell suspension of the antagonistic yeast strains and VOCs produced by the antagonistic yeast strains on postharvest mango fruits at 25 °C for 7 days.

Treatments	Yeast Cells		VOCs	
	Wound Diameter (mm) ^a^	Disease Severity Reduction (%) ^b^	Wound Diameter (mm) ^a^	Disease Severity Reduction (%) ^b^
Fruit Rot				
*L. theobromae* (positive control)	50.7 ± 1.37	0	57.2 ± 5.30	0
*L. theobromae* + *T. indica* DMKU-RP31	24.2 ± 1.74	52.7 ± 2.66b	39.2 ± 5.56	31.5 ± 0.97b
*L. theobromae* + *T. indica* DMKU-RP35	8.5 ± 0.16	82.4 ± 5.64a	26.8 ± 2.08	53.1 ± 1.36a
*L. theobromae* + *Ps. hubeiensis* YE-21	20.2 ± 1.09	49.5 ± 3.89b	46.2 ± 2.97	19.2 ± 1.52c
*L. theobromae* + Benomyl	8.3 ± 0.55	87.5 ± 4.32a	nd	nd
Anthracnose				
*C. gloeosporiodes* (positive control)	29.6 ± 0.68	0	57.0 ± 4.73	0
*C. gloeosporiodes + P. aspenensis* DMKU-SP67	1.7 ± 0.53	94.1 ± 0.18a	29.3 ± 0.88	48.5 ± 0.15
*C. gloeosporiodes +* Benomyl	1.7 ± 0.42	93.9 ± 3.32a	nd	nd

nd: not determined. Data in the same column data followed by the different, same, and overlapping lowercase letters means significantly different and not significantly different according to the overlapping of Duncan’s multiple range test at *p* ≤ 0.05. Each result presents the mean ± standard derivation from three replicates. ^a^ Wound diameter was an average wound diameter from five mango fruits; each of the wound diameters were measured and averaged. ^b^ The disease severity reduction (%) = (diameter of the wounds on the positive control − diameter of the wounds on the treatment/diameter of the wounds on the positive control) × 100.

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
