# Peer review of "Biological Control of Fruit Rot and Anthracnose of Postharvest Mango by Antagonistic Yeasts from Economic Crops Leaves"

_microorganisms, 2020, doi:10.3390/microorganisms8030317_

Round 1

Reviewer 1 Report

The authors present some interesting and valuable date on the biological control of two diseases of the important crop fruit mango. The manuscript is well written; materials and methods are adequate and the results are presented clearly. Only four out of 307 yeast strains showed some efficacy against postharvest disease, with limited inhibition of the pathogenic fungal strains both in Petri dishes as well as on fruits. The authors could give some additional information on the four yeast strains, and discuss possibilities of improving efficacy against the mango pathogens, maybe by combining them or by developing integrated pest management options using bio-control agents in addition to low doses of pesticides etc. The next research plans could be discussed, as well as possibilities to commercialize the BCAs.

Reviewer 2 Report

The manuscript "Biological Control of Fruit Rot and Anthracnose of Postharvest mango by Antagonistic Yeasts from Economic Crops Leaves" reports on the interesting finds of 4 antagonistic fungi that appear to have potential to inhibit the causative agents of end rot and anthracnose.  The experimental approach was sound the the results were interesting.

The authors need to address a number of points - outlined below:

Grammatical Points:

Line 33 important fruits (should be fruit)

Line 51 producing, not produced

Line 58 – this is an incomplete sentence:  However, due to growing concerns about the potential risks fungicides pose for human health, environmental contamination,

and the development of fungicide resistance by pathogens [8].

Line 80 – use of antagonistic the second time is redundant.

Line 165 – Change Evaluation to Evaluating

Line 203 – data = plural, These data not the data.

Line 207 – pathogens cause causing of……Something wrong here – not certain what you are saying

Line 282 starting with Evaluation the efficiency…. This is an incomplete sentence with no meaning.  Please clarify.

Line 303 – Change to Evaluation of the efficacy…..

General Points:

What was the rationale to screen important economical crops of Thailand rather than screening for yeast found on Mango? 

How will the expected biocontrol agents affect the quality, i.e. taste, of the mangos?

Format of figure 3 needs to be cleaned up.  Use solid bars rather than patterns.  Remove shadow affect from bars.  Remove background lines.  Make X-axis a solid black line

Add a figure showing typical Siderophore detection results.
